# Profile and Different Approaches for Size Characterization of Microplastics in Drinking Water from the Lisbon Water Supply System

**DOI:** 10.3390/molecules29184426

**Published:** 2024-09-18

**Authors:** Rodrigo D. M. Cordeiro, Vitor V. Cardoso, Rui N. Carneiro, Cristina M. M. Almeida

**Affiliations:** 1Direção de Laboratórios, Empresa Portuguesa das Águas Livres, S.A.—EPAL, 1250-144 Lisboa, Portugal; rodrigo.cordeiro@adp.pt (R.D.M.C.); vitor.cardoso@adp.pt (V.V.C.); rui.carneiro@adp.pt (R.N.C.); 2Laboratório de Bromatologia e Qualidade da Água, Faculdade de Farmácia da Universidade de Lisboa, 1649-003 Lisboa, Portugal; 3iMed. UL, Faculdade de Farmácia da Universidade de Lisboa, 1649-003 Lisboa, Portugal

**Keywords:** microplastics, drinking water, micro-FTIR, microplastic characterization, polymer identification

## Abstract

Microplastics (MPs) contribute to the overall pollution of water sources, affecting not only aquatic ecosystems but also water for human consumption (WHC). Currently, there needs to be a global consensus on safe levels of microplastics in WHC, which will allow regulatory efforts and risk assessments to be carried out. Therefore, this study aims to characterize MP particles in WHC of the Lisbon water supply system (LWSS) and compare two approaches to quantify these particles (length and width of the particles, and the area equivalent diameter (AED) of the particles). The quantification of MP particles was made via micro-FTIR (Fourier Transform Infrared Spectroscopy) on transmission mode after water filtration on 5 µm silicon filters. Thirty-eight WHC samples from the LWSS showed MPs up to 836 MPs/L, with an average value of 196 MPs/L. The most representative polymer was polyethylene (PE, 77.2%). The other eight polymers were also quantified. The length and width of MPs ranged between 84 µm and 41 µm, respectively. The AED of MPs ranged between 24 µm and 405 µm. The MP dimensions of both approaches can differ significantly.

## 1. Introduction

Globalization has made modern society dependent on plastics, which became significant in the 50s. Since then, plastic products have quickly become indispensable daily [1].

Plastic is any synthetic or semi-synthetic polymer with thermoplastic or thermo-rigid properties, synthesized from non-renewable raw materials such as hydrocarbons or biomass [2]. This polymeric material has several advantages: versatility, stability, durability, safety, lightness, and, above all, low production cost [1]. This wide range of properties gives this material great applicability in various areas, including the textile, electronics, automotive, food, and medical industries [2,3]. However, the inappropriate use and disposal of plastics, rapid diffusion, excessive production, and the slow process of their degradation increase the content of plastic waste, which, when released into the environment, breaks down into smaller plastics, leading to microplastics (MPs). The adverse effects of these can be severe, affecting both the environment and potentially human health [4,5]. Addressing this issue requires improved detection methods, effective removal technologies, stringent regulations, and comprehensive monitoring efforts to mitigate the potential impacts on human health and the environment.

MPs are defined as any solid synthetic or polymeric matrix particle between 1 μm and 5 mm in size. They are also very heterogeneous particles, encompassing various origins, shapes, sizes, colors, polymer types, and physicochemical properties [6,7].

Furthermore, MPs can be categorized into two broad categories: primary and secondary. Primary microplastics are small particles that are intentionally used, such as in personal care products (cosmetics, scrubs). On the other hand, secondary microplastics are derived from the degradation and deterioration of macroplastics or primary microplastics caused by UV radiation, mechanical action, abrasive physical processes, microbiological action, or other natural degradation processes [6,7].

Many publications show that more than 80% of waste in natural waters, especially freshwater, is plastic [8,9,10,11]. The main objects listed are reusable bags, straws, fabrics, and packaging for cleaning and hygiene items (mainly composed of PE), PET water bottles, PP bottle caps and food containers, PS styrofoam and disposable products, and PA ropes and fishing nets. These plastic items can fragment and generate various MP particles through erosion and oxidation in these waters. Due to their persistence and ubiquity, these particles appear in drinking water, as treatment systems are ineffective at removing them. The inevitability of microplastics in the water supply system is a direct consequence, and it is crucial to assess the unintentional occurrence of microplastics in natural freshwaters intended to produce water for human consumption (WHC) [1].

Despite the growing international concern surrounding MPs, more effective legislation is currently needed to restrict their use or define guideline values in environmental matrices and drinking water [8].

Regarding the microplastic particles identified in aquatic environments, the most representative polymers are polyethylene (PE), polypropylene (PP), polystyrene (PS), polyvinyl chloride (PVC), polyethylene terephthalate (PET), and polyamide (PA) [12].

The environmental effects of MPs are substantial, given the high persistence of these microparticles in the environment, particularly in raw natural waters (surface and groundwater). They can disrupt the ecological balance of the ecosystem and have a negative impact on biodiversity [13,14].

However, based on the limited evidence available, there are currently no relevant data to suggest that there is a direct risk to human health associated with exposure to MPs through WHC. Nevertheless, regardless of whether there are risks to human health from exposure to these particles, it is crucial to improve plastics management and reduce the amount of waste and pollution to protect the environment and human well-being [8]. 

In addition to all these shortcomings, the monitoring of MPs in water matrices is still minimal due to the lack of harmonized analytical methodologies, especially due to the lack of standardization of laboratory conditions, sampling, sample treatment, filtration, data acquisition and processing, and method optimization and validation. As such, the need to define the most appropriate methodologies for monitoring microplastics in water matrices is pressing and has been the main concern of international institutions, particularly European ones, to ensure that the risk of microplastics to human health and the environment is assessed and that the occurrence of these particles is evaluated [15].

An example of the work carried out is reflected in Directive (EU) 2020/2184 [16], transposed into Portuguese national legislation by Decree-Law 69/2023 [17], which defines, among other requirements, that materials in contact with water must not release contaminants into the water at levels higher than necessary, considering the intended purpose of the material. 

Although several methodologies may show promising results when analyzing microplastics, the most recommended are vibrational spectroscopic methods coupled with optical microscopy, particularly Raman microscopy (µ-Raman or micro-Raman) and microscopy associated with Fourier transform infrared spectroscopy (µ-FTIR or micro-FTIR) [18]. 

Due to the inconclusive results and the lack of relevant information, MPs are currently considered contaminants of emerging concern. Additional details are needed concerning the origins of the contamination, its effects on the environment and human health, and the suggested analytical approach to be employed [15,19].

In addition, the dimensions of MPs reported in studies can vary significantly based on the methods used for their quantification and approaches use for MPs size characterization. Filtration pore sizes, analytical techniques, and quantification approaches all play a role in determining the size distribution of detected MPs. Understanding these methodological differences is crucial for comparing studies and developing standardized protocols for microplastic research. In this sense, this work aims to continue the studies on the occurrence of microplastics that have already begun in the Lisbon distribution network [20], as well as to compare the size of the MPs in the samples analyzed using two different approaches: (i) determining the length and width of the particles, and (ii) determining the area equivalent diameter (AED) of the particles.

The differences in size characterization methods of microplastic particles can have significant implications for regulatory decisions, ecological risk assessments, and the comparability of results across different studies [21].

Regulatory bodies rely on consistent size thresholds to manage microplastic pollution, but variations in these methods can lead to differing interpretations of what constitutes a microplastic, affecting regulatory standards and the enforcement of environmental legislation. Inconsistent characterization of PM size between regions can also lead to inconsistent regulations, making it difficult to align international policies [22].

In ecological risk assessments, the size of microplastic particles influences their environmental behavior and interaction with organisms, affecting exposure, bioavailability, and toxicity. Larger particles may be less ingestible by smaller organisms, while smaller particles can penetrate biological barriers more easily, leading to varying toxicological outcomes. Differences in size characterization methods can therefore lead to inconsistent conclusions about the ecological risks posed by microplastics, potentially skewing risk assessments [23].

Furthermore, variations in size characterization methods hinder the comparability of results across studies, making it challenging to synthesize findings and draw broad conclusions about the environmental and health impacts of microplastics. This inconsistency complicates meta-analyses and systematic reviews, as the lack of standardization may force researchers to exclude studies or apply correction factors, introducing further uncertainties into their analyses [21,22,23].

To address the problem of the characterization of MP particles, a micro-FTIR method in transmission mode was previously validated [20] and applied to monitor several target polymers in WHC samples from LWSS: polypropylene (PP), polyethylene (PE), polyvinyl chloride (PVC), polyethylene terephthalate (PET), polystyrene (PS), polyurethane (PUR), polytetrafluoroethylene (PTFE), polyvinylidene fluoride (PVDF), polymethyl methacrylate (PMMA), polycarbonate (PC), ethylene propylene diene monomer (EPDM), and polyamide (PA).

This study was performed in EPAL (Empresa Portuguesa das Águas Livres, SA), the most prominent national company in the water supply industry and a significant part of Grupo Águas de Portugal (AdP). The company is involved in water supply and sanitation and has a significant impact on the environmental sector in Portugal.

## 2. Results and Discussion

### 2.1. Polymer Profile

The distribution of MP particles in WHC can be grouped by the total number of MP particles of each polymer relative to the total number of MP particles of the water samples (Figure 1) or by the polymer frequency in target water samples (Table 1). Table 1 also shows the total of MP particles found in WHC samples from LWSS organized by minimum (Min), maximum (Max), and median (Med) of the number of MP particles. The number of positive samples (Pos) and frequency (Freq) were also reported. The Freq equals the number of positive samples divided by the number of studied water samples. 

The most common polymer particles were PE (77.2%), PET (10.9%), PA (3.8%), PP (3.6%), PS (2.3%), EPDM (1.4%), PTFE (0.3%), PUR (0.3%), and PMMA (0.2%) (Figure 1). This representativeness follows the previous results obtained in 2023 in LWSS [17]. 

This order also aligns with the global profile of the most synthesized plastics and the trend of major contaminating macroplastics or large plastic fragments identified in raw natural water intended for drinking water production [24].

Relative to their frequency in the target water samples, the most representative polymers were PE (100%), PET (82%), PP (58%), PA (49%), and PS (39%). The remaining polymers showed a frequency lower than 20%.

Two polymers showed a frequency higher than 80%, namely PE (100%) and PET (82%). The number of different polymers per water samples ranged between 2 and 8, with a median of 4 different polymers per water sample.

### 2.2. Overview of MPs in LWSS

Figure 2 shows the results of the dispersion of MP concentrations in WHC at fourteen sampling points (A1-L2) of the EPAL water supply system. 

The average concentration of MPs identified in WHC in each sampling point ranged from 48 MPs/L (K) to 550 MPs/L (E), with an average of 215 MPs/L and a median of 155 MPs/L. 

Of the 14 sampling points, 2 had MP particles below 100 MPs/L, 6 sampling points had MP particles between 100 and 200 MPs/L and 6 sampling points had a higher number of particles (>200 MP/L).

The dispersion of values is very high between sampling points, even when the water pipes come from the same water storage tank, such as samples A1 and A2 and L1 and L2.

The number of MP particles can double at some sampling points (E, I and H), especially for PE. The number of MP particles of the other polymers varied between 2 MPs/L (PTFE and PMMA) and 29 MPs/L (PTFE), having mean amounts ranging from 5 MPs/L (PMMA) to 8 MPs/L (PTFE), and with relative standard deviations between 32.9% (PUR) and 93.3% (PTFE).

Sampling points A2, E, F and I showed the highest content of MP particles with 309, 550, 383, and 470 MPs/L, respectively. PE is the most representative polymer in any of these WHC samples.

Most of the studies on MPs and water contamination focus on marine and freshwaters and biota. There are few results on the occurrence of these particles in WHC. Therefore, any comparison is minimal. The findings, however, closely resembled the outcomes reported by Tong H. et al. [25], who demonstrated a high presence of MPs in drinking water in China, ranging from 0 to 1247 MPs/L, with an average concentration of 440 ± 275 MPs/L. In contrast, Kosuth M. et al. [26] documented a significantly lower MP content in drinking water samples compared to those in our study, with levels varying from 0 to 61 MPs/L and an overall average of five particles/L. The profile and number of polymer particles are consistent with our previous results obtained in 2023 [20]. Although the samples belong to the same water distribution network, the sampling points are different and are consequently affected by the distribution network itself (constitution, size, and aging).

### 2.3. Spectral Similarity

Table 2 shows the spectral similarity (match %) of the spectra obtained from the libraries. 

A high spectral similarity index was obtained for all the polymers identified, with values between 65% (PE, PET, PA, PP, and PS) and 96% (PP). The mean and median values of the spectral similarity index were equal and ranged between 72% (PET) and 83% (PUR). Therefore, based on the values obtained from the spectral similarity index, it can be concluded that the identification of the different PM particles in the water samples was successful.

### 2.4. Polymer Size

Figure 3 and Figure 4 show the results of the variation in the size of the MP particles identified in the WHC samples and the variation of the number of MP particles by 10-dimension groups.

MP particles identified in WHC samples showed an average length and width of 84 µm and 41 µm, respectively. PE, PET, and PA showed the highest variation in the length of MP particles with dimensions up to 641 µm (Figure 3).

In all, 58.7% of MP particles from different polymers are between 20 and 150 µm in size. The remaining MP particles are evenly distributed among the other size groups, except for the largest particle group (>500 µm), which is negligible (0.03%) (Figure 2). Only PS showed a higher number of smaller particles (20–150 µm), twice as many as the other polymers.

The dimensions of MPs can vary significantly depending on the approach used for their quantification. Different methods and techniques for sampling, analyzing, and quantifying microplastics can yield different size distributions and concentrations. Understanding these methodological differences is crucial for comparing studies and developing standardized protocols for microplastic research. Regarding the MP dimensions, there are various ways of presenting the results: particle length and width, particle area, and area equivalent diameter (AED). Some of these approaches are inherent to the quantification method and the software associated with the equipment, the most common being particle length and width. However, some regulatory documents adopt different ways of quantifying particle size, namely the AED [27].

Figure 5 shows the MP particle sizes expressed in AED. The size profile is very similar to that of the length approach (Figure 3), but the size of the polymer can differ significantly. The AED of MP particles identified in WHC samples ranged between 24 µm (PE) and 405 µm (PUR), with a median AED between 37 µm and 107 µm.

Figure 6 compares each polymer’s median MP particle size with the median AED for the same polymer to better understand their differences.

No correlation exists between the length and AED of MP particles (R^2^ = 0.2182).

Only three polymers show deviations lower than or equal to 10% between the two approaches to quantifying particle size (length and AED), namely PP (8.7%), PS (10.3%) and PMMA (−1.5%). The deviations obtained in some polymers are much greater between the two approaches. Regarding the MPs length, it is higher for PP (16.5%), PET (21.2%), PA (17%), and EPDM (51.8%). For PTFE and PUR, the dimensions are higher in AED, namely 38.4% and 18.2%, respectively.

These differences are not associated with the type of polymer but with the irregularity of the particle. Other samples may show different results for the same polymers. It is important to emphasize that the two approaches can produce very different results, which has regulatory implications for the interpretation of interlaboratory tests and the assessment of the potential impact on ecosystems and human health.

To determine if there is a significant difference between these two approaches, a Mann–Whitney U test was applied. There was a significant difference in MP particle dimensions between sizes that were quantified by AED and sizes based on length; z = −8.004, *p* = 1.21 × 10^−15^. Since the *p*-value is lower than 0.05, the null hypothesis was rejected. Therefore, there is sufficient evidence to say that the true mean sizes are different between the two groups. The MP particles size is higher for length’s approach.

The effect size (*r*) was 0.196. Therefore, a small effect of sample size of groups was obtained.

Studies should clearly report the methods used for size characterization, including the size ranges considered and the techniques employed. This transparency allows for a better comparison across studies and helps to identify potential sources of variability in findings.

Continued development and validation of size characterization methods are crucial to ensure they accurately reflect the environmental reality and biological relevance of microplastic pollution.

## 3. Materials and Methods

### 3.1. Study Area

EPAL manages the Lisbon water supply system, which collects, treats, transports, and supplies water to around 2.9 million people in thirty-three municipalities in the Greater Lisbon region. Monitoring water quality throughout EPAL’s supply system, from the water resources to the point of delivery to the consumer, is one of the company’s major concerns. 

As such, the company’s policy involves the development of analytical methods that allow the search for various non-legislated compounds, thus enabling the characterization of the water at its source, the assessment of the products used in the water treatment plants and periodic monitoring of these compounds in WHC. 

The drinking water journey from the water treatment plant outlet to the storage tanks in Lisbon’s distribution network covers several kilometers of pipes and ducts, passing through various materials in the supply system. Lisbon’s distribution network consists mainly of high-density polyethylene (HDPE), polyethylene (PE), ductile iron, fiber cement, cast iron, and reinforced concrete. Polymeric materials, particularly HDPE and polyvinyl chloride (PVC), are commonly used for pipe manufacturing [20]. 

Surface waters undergo treatment at the water treatment plant and are then stored in a water storage tank before being distributed. EPAL’s water treatment plants have tanks with a tap for collecting grab samples of treated water. Throughout the Lisbon supply system, all storage tanks are equipped with a tap for collecting grab samples of water.

The study area included the outlet of two WTPs, Asseiceira WTP (L1 and L2, located in Tomar) and Vale da Pedra WTP (A1 and A2, located in Cartaxo), and ten water storage tanks located in Lisbon (B, C, D, E, F, G, H, I, J, K). 

### 3.2. Sampling

The WHC samples were gathered in 1000 mL amber glass containers under the following conditions [20]: (i) Without the use of preservatives; (ii) Ensuring that there is at least a 2 min tap flushing before collecting samples; (iii) Rinsing the container with the water to be analyzed before collecting the water; (iv) Preventing any contact or contamination of the bottle’s neck, cap, or inside; (v) Filling the bottle to approximately 1 cm from the neck (near the top) and shaking it. 

Between January and June 2024, 38 WHC samples were gathered across EPAL’s distribution network. 

Most of the samples were obtained from 14 sampling locations in Lisbon. The water samples were stored in dark conditions at 5 ± 3 °C and refrigerated, with a maximum storage period of 7 days before analysis [28,29].

### 3.3. µ-FTIR Analysis

The analytical method used in this study to quantify MP particles in WHC had already been implemented and validated: 250 mL of water sample (without sample pre-treatment) was filtered through 5 µm silicon filters and analyzed by Fourier transform infrared spectroscopy method coupled with optical microscopy (micro-FTIR) in transmission mode. The filtration step was performed in a laminar flow chamber. The mapping was performed under the following conditions: OMNIC correlation, use of reference and in-house libraries, silicon filter background, and 100 µm × 100 µm aperture. The validated method is accurate, with an average recovery of 91% and a coefficient of variation of 14%. The validated method shows a reporting limit (RL) of 44 MPs/L [20]. 

MP particles were controlled on blanks (mineral water) and contamination was avoided for every sample. Positive controls with different polymers and sizes were used. Only spectra with a percentage of spectra similarity or hit quality index (HQI), also known as match %, set to at least 65% of recognition, were accepted [30,31]. 

Polymer reference libraries supplied by Thermo Scientific (“Biblioteque Particules”, “Polimeri”, “Wizard Poly”, “HR Sprouse Polymers by Transmission”, “Test micropol”, and “Aldrich FT-IR Collection Edition II”) and homemade spectral libraries were used to identify the polymers. 

MP sizes (length and width) were collected using the Wizard section. Micro-FTIR allowed measuring or estimating the number of MP particles higher or equal to 20 µm on the full filter or sample support. When the total number of particles on the filter was too high to measure in a practical time, smaller sub-areas of the filter were used. The sub-sampling areas covered at least 20% of the area of the sample filter [30]. 

The dimensions of MP particles were assessed by length, width and AED.

The AED means the diameter of a circle having the same area as the 2-dimensional projection of the particle’s optical or hyperspectral chemical images (Figure 7) [27]. 

For this determination, the area (A) of each particle is calculated from the length (L) and width (W) provided by the equipment. From this area, the AED is determined, according to Equation (1).
(1)$$AED=2\times \sqrt{\frac{A}{\pi }}$$

### 3.4. Statistical Analysis

All statistical tests were performed using Microsoft Excel software. Basic descriptive statistics were applied to evaluate MP particles in water samples (average, median, minimum, maximum). Boxplot charts were also used for the comparison of MP dimensions between polymers. For the comparison of differences between MP dimensions by both approaches and because the data do not follow a normal distribution, the Mann–Whitney U test was applied because only a non-parametric statistical comparison among groups is adequate. Therefore, the Mann–Whitney U test was used to determine if there is a significant difference in MP particle dimensions between the AED and length approach. A sample size of 1.676 (total of MP particles of all polymers quantified) is well within the range for which the Mann–Whitney U Test is appropriate. The test is feasible and robust at this scale, providing reliable results for detecting group differences. To evaluate the effect size (r) in the Mann–Whitney U Test, Equation 2 was applied [32]:(2)$$r=\frac{\left|z\right|}{\sqrt{n}}$$
where z is z-value of z-distribution and n is the number of MP particles analyzed in each approach.

Regarding the effect strength, *r* less than 0.3 is a small effect, *r* between 0.3 and 0.5 is a medium effect, and an *r* greater than 0.5 is a large effect [32]. 

## 4. Conclusions

The micro-FTIR method allowed the identification of MP particles and their dimensions.The presence of MP particles was confirmed in water for human consumption of Lisbon’s water supply system.The number of MP particles in WHC in each sampling point ranged between 20 MPs/L and 836 MPs/L. The average concentration ranged from 48 MPs/L to 550 MPs/L with an average of 215 MPs/L and a median of 155 MPs/L.The heterogeneity of the type and size of the polymers is high.The profile of MP particles is similar in all sampling points.Some sampling points have double the number of particles.MP particles belong to nine different polymers, with PE being the most abundant.MP particle dimensions ranged between 20 and 641 µm, with an average length and width of 84 µm and 41 µm, respectively.PE, PET, and PA showed the highest variation in the length of MP particles.There is no correlation between length and AED.Further research should focus on how the raw waters may produce WHC in order to comprehend better the pathways and sources of MP particles in the water supply system.Significant advances in MP research can be expected to obtain harmonized data acquisition and reporting protocols. The methods used to characterize the size of microplastic particles play a critical role in shaping regulatory decisions, assessing ecological risks, and ensuring the comparability of scientific findings. Standardization and transparency in these methods are essential to address the challenges posed by microplastic contamination.The obtained results allow us to reconsider the materials currently employed in the water distribution networks and define strategies to prevent or manage the identified risks, which will ensure the safety of drinking water.

## Figures and Tables

**Figure 1 molecules-29-04426-f001:**
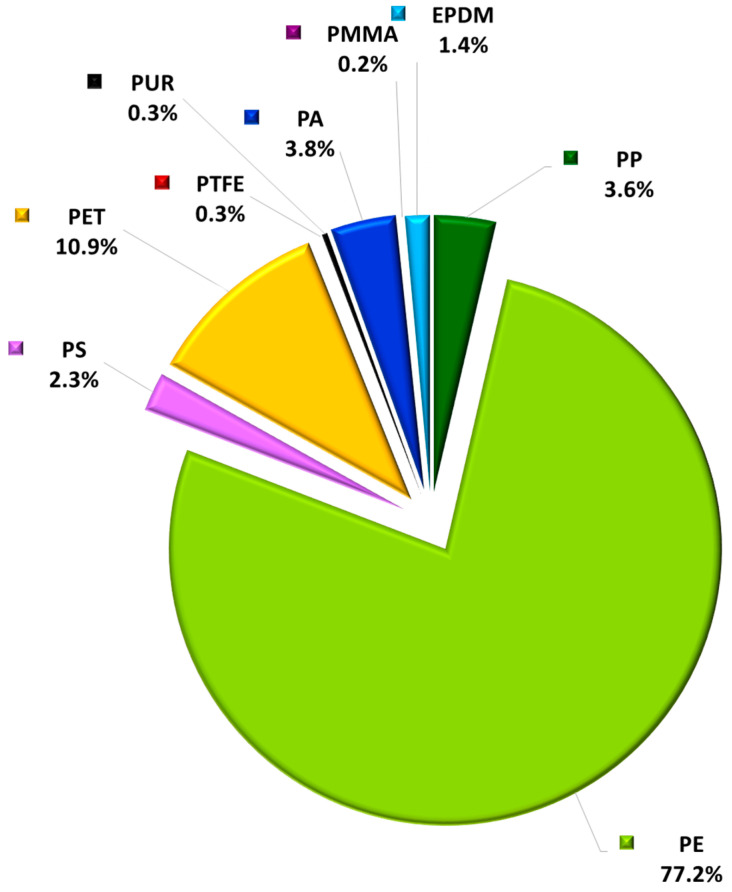
Profile and distribution of MP particles in WHC from Lisbon water supply system of EPAL expressed as a percentage of the total number of MP particles.

**Figure 2 molecules-29-04426-f002:**
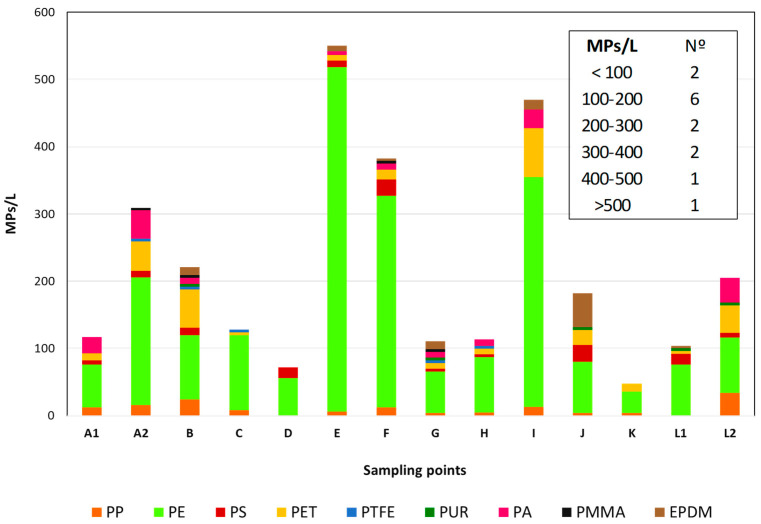
Average of MP particles in water for human consumption at 14 Lisbon water supply system sampling points.

**Figure 3 molecules-29-04426-f003:**
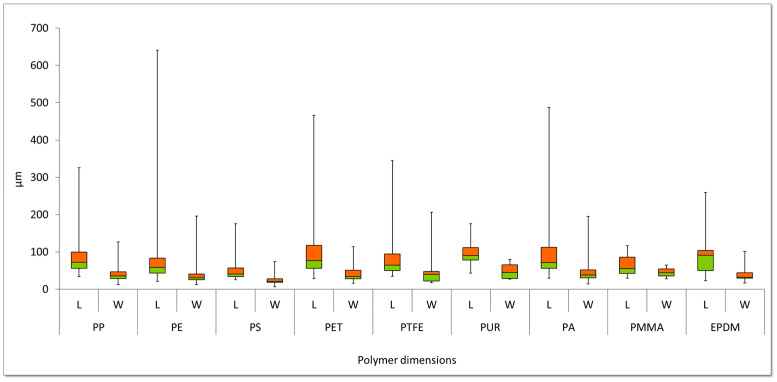
Dimensions, length (L) and width (W) of MP particles in WHC from EPAL’s supply system (n = 38).

**Figure 4 molecules-29-04426-f004:**
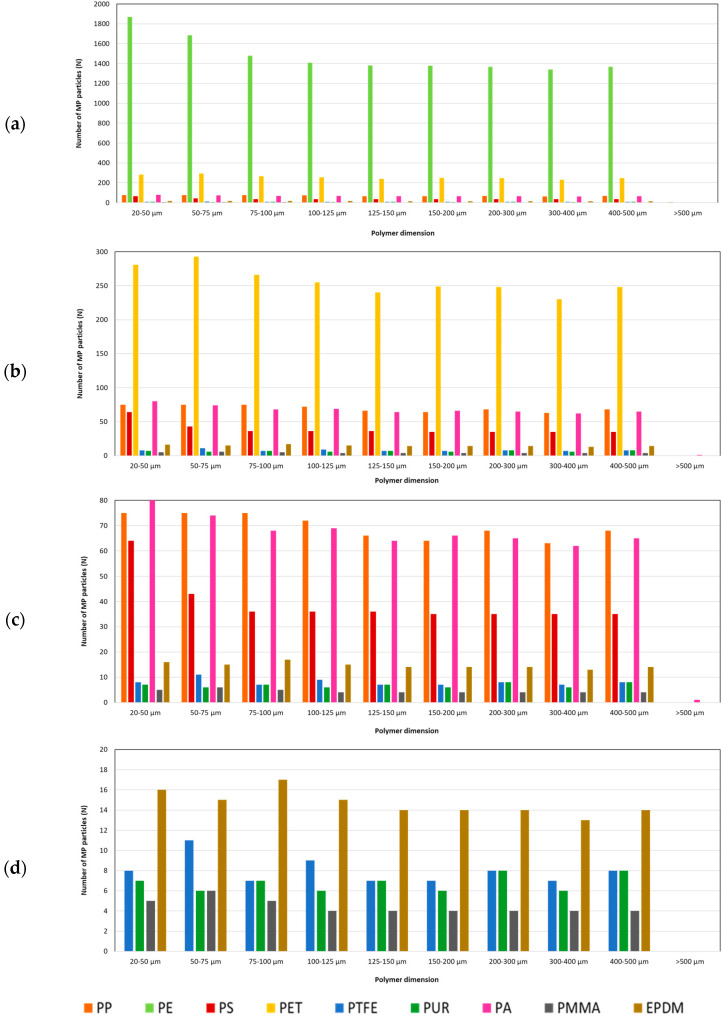
Distribution of polymer particles by their sizes in all samples of WHC from the Lisbon water supply system between January and June 2024: (**a**) up to 2000 MP particles; (**b**) up to 300 MP particles; (**c**) up to 80 MP particles; and (**d**) up to 20 MP particles (n = 38).

**Figure 5 molecules-29-04426-f005:**
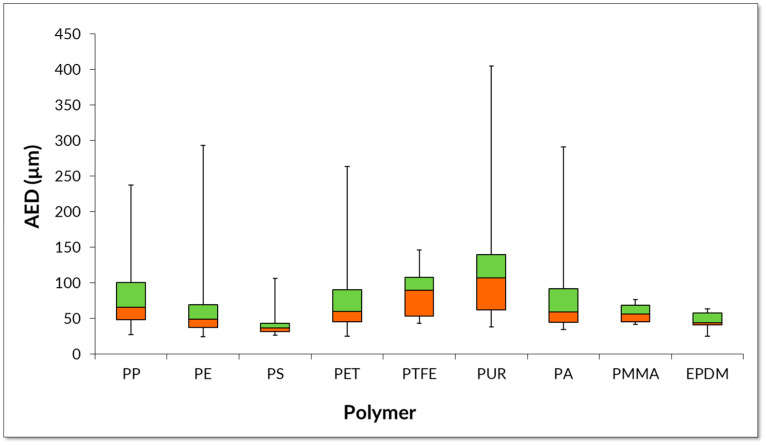
Dimensions of MP particles in WHC from EPAL’s supply system expressed in area equivalent diameter (AED) (n = 38).

**Figure 6 molecules-29-04426-f006:**
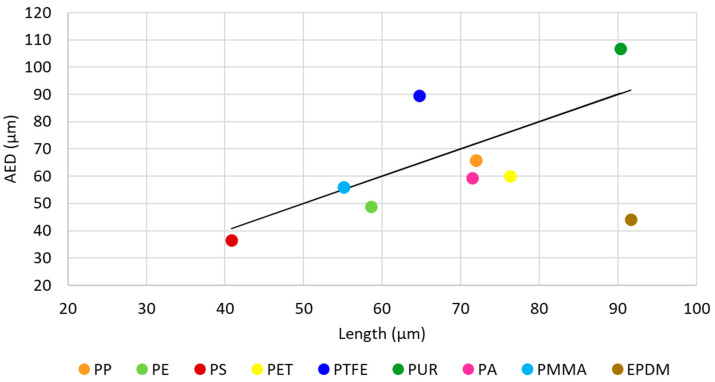
Comparison of the median of MP particle dimensions in WHC from EPAL’s supply system quantified by two approaches: length (L) *versus* area equivalent diameter (AED).

**Figure 7 molecules-29-04426-f007:**
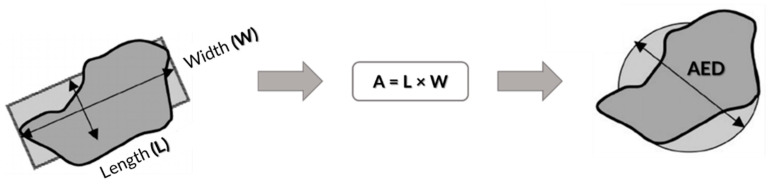
Method for determining particle size by area equivalent diameter (AED) in µm.

**Table 1 molecules-29-04426-t001:** MP particle distribution (MPs/L) in water for human consumption in the Lisbon water supply system (n = 38).

	PE	PET	PA	PP	PS	EPDM	PTFE	PUR	PMMA
Min	16	4	4	4	4	4	4	4	4
Max	828	137	68	60	25	51	4	4	4
Med	96	12	10	7	8	12	4	4	4
Pos	38	31	18	22	15	7	5	6	4
Freq (%)	100	82	47	58	39	18	13	16	11

**Table 2 molecules-29-04426-t002:** Percentual values of hit quality index or match of MP particles in WHC from LWSS. Number of MP particles (n), minimum (Min), maximum (Max), median (Med), and average (Mean) of obtained match values.

	PE	PET	PA	PP	PS	EPDM	PTFE	PUR	PMMA
n	1400	227	59	61	36	13	7	6	5
Min	65	65	65	65	65	66	69	73	66
Max	96	89	91	96	93	88	89	89	91
Med	81	72	77	80	72	70	82	83	74
Mean	81	72	77	80	72	71	80	83	75

## Data Availability

The data on MP particles in water for human consumption are private due to the identification of sampling points.

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
