# Peer review of "Profile and Different Approaches for Size Characterization of Microplastics in Drinking Water from the Lisbon Water Supply System"

_molecules, 2024, doi:10.3390/molecules29184426_

Round 1
Reviewer 1 Report
Comments and Suggestions for Authors
1. The title should highlight the source and characteristics of the sample;
2. the authors mention quantification by two different methods, but only one, FTIR, is seen in the abstract;
3. the authors introduce too many basics in the introduction chapter that are not relevant to the topic. However until reading this section I didn't see which two methods the authors used for comparison, Raman spectroscopy and FTIR?
4. The Results and Discussion section is an entirely typical monitoring study. I don't see where the methodology has been improved.
Author Response
Comments and response to Reviewer 1: 1. The title should highlight the source and characteristics of the sample; We agree with this comment. Therefore, the title was changed to “Profile and different approaches for size characterization of microplastics in water for human consumption from Lisbon” 2. the authors mention quantification by two different methods, but only one, FTIR, is seen in the abstract; The MP particles were analysed by FTIR method as described in the abstract and in the methods. However, the dimensions of MP particles were evaluated by two approaches: i) determining the length and width of the particles, and ii) determining the area equivalent diameter (AED) of the particles. This information is in lines 110-112.
3. the authors introduce too many basics in the introduction chapter that are not relevant to the topic. However until reading this section I didn't see which two methods the authors used for comparison, Raman spectroscopy and FTIR? The MP particles were analysed by micro-FTIR in transmission mode. Two approaches were used to characterize their size. The introduction was carefully revised to reinforce this point.
4. The Results and Discussion section is an entirely typical monitoring study. I don't see where the methodology has been improved.
The scope and objective of this work are not to improve methods for analyzing microplastics in water. The most used methods are vibrational spectroscopic methods coupled with optical microscopy, particularly Raman microscopy (µ-Raman or micro-Raman) and microscopy associated with Fourier transform infrared spectroscopy (µ-FTIR or micro-FTIR). In this study, we only used the micro-FTIR method. However, regardless of the optimization and validation of the methods, it is necessary to consider the particle quantification methods. Otherwise, we could be considering very different dimensions for the same particle. This approach is relevant as it is often omitted from published results. Therefore, this paper compares the dimensions of the MP particles by these two approaches.
|
Reviewer 2 Report
Comments and Suggestions for Authors
Title: Profile and different approaches for size characterization of microplastics in water samples
The manuscript evaluates and compares various methodologies for size characterization of microplastics (MPs) in water samples, highlighting differences in accuracy, precision, and applicability among techniques such as micro-FTIR in transmission mode. The study aims to address the inconsistencies in results across different approaches and proposes guidelines for standardizing MP size characterization to improve comparability in future research. However, several areas need improvement to enhance the paper's clarity, completeness, and impact.
1. The manuscript presents data on MP particle sizes and distributions using different quantification methods (length and width vs. area equivalent diameter), but it lacks detailed statistical analysis to validate the findings.
Please incorporate robust statistical analyses, including confidence intervals, standard deviations, and hypothesis testing, to support the differences observed between the two methods. This will provide a stronger basis for the conclusions and highlight the significance of the methodological differences.
2. While the study compares two approaches for MP size characterization, it does not thoroughly discuss the implications of these differences on regulatory assessments, environmental impact evaluations, and inter-laboratory comparisons.
Please expand the discussion to include a detailed analysis of how the differences in size characterization methods could affect regulatory decisions, ecological risk assessments, and the comparability of results across different studies. Provide specific examples or case studies to illustrate these points.
3. The introduction is comprehensive but could better emphasize the study's practical implications and the necessity for standardized methodologies in MP research.
4. The conclusion should succinctly summarize the main findings and their implications. It should also provide clear recommendations for future research and regulatory considerations.
5. On page 6, Figure 3, what does the orange and green color mean in the chart? Please annotate the diagram to make it more transparent.
Comments on the Quality of English LanguageExtensive editing of English language required.
Author Response
Comments 2: The manuscript evaluates and compares various methodologies for size characterization of microplastics (MPs) in water samples, highlighting differences in accuracy, precision, and applicability among techniques such as micro-FTIR in transmission mode. The study aims to address the inconsistencies in results across different approaches and proposes guidelines for standardizing MP size characterization to improve comparability in future research. However, several areas need improvement to enhance the paper's clarity, completeness, and impact.
1. The manuscript presents data on MP particle sizes and distributions using different quantification methods (length and width vs. area equivalent diameter), but it lacks detailed statistical analysis to validate the findings. Please incorporate robust statistical analyses, including confidence intervals, standard deviations, and hypothesis testing, to support the differences observed between the two methods. This will provide a stronger basis for the conclusions and highlight the significance of the methodological differences. Thank you for pointing this out. We agree with this comment. A new section about statistical analysis was included (lines 376-394) and the discussion of obtained results (lines 285-299).
2. While the study compares two approaches for MP size characterization, it does not thoroughly discuss the implications of these differences on regulatory assessments, environmental impact evaluations, and inter-laboratory comparisons.
Please expand the discussion to include a detailed analysis of how the differences in size characterization methods could affect regulatory decisions, ecological risk assessments, and the comparability of results across different studies. Provide specific examples or case studies to illustrate these points.
We agree with this comment. The discussion section was carefully revised.
3. The introduction is comprehensive but could better emphasize the study's practical implications and the necessity for standardized methodologies in MP research. The introduction was carefully revised. There are new sentences in the revised manuscript.
4. The conclusion should succinctly summarize the main findings and their implications. It should also provide clear recommendations for future research and regulatory considerations. Thank you for pointing this out. We agree with this comment. The conclusions have new sentences (lines 411-422). 5. On page 6, Figure 3, what does the orange and green color mean in the chart? Please annotate the diagram to make it more transparent. It is a boxplot chart or whisker plot; the color is unimportant, but it facilitates the identification of the median, minimum, maximum, first, and third quartile. Usually, these charts are in two colors, but each color does not represent any values.
|

Reviewer 3 Report
Comments and Suggestions for Authors
Dear authors,
I appreciate your work, I believe the item treated is very rilevant nowadays.
I have some minor concerns:
- paragraph 2.1, line 143 you wrote: PE and PET showed a frequency higher than 80%, 100%, and 82% respectively. You refer to 2 polymer but than report 3 values?
- paragraph 2.3, line 186 you stated that "it can be concluded the various particles in the samples analyzed were identified well". Was this conclusion derived from the spectra similarity index? Or do you use some other measuring method?
- figure 4, the brown lines to which polymer refer? There isn't any indication in the label of the figure for this color.
- In the conclusion, I would report at least a range (minimum and maximum values registered) of the number of particles detected with their errors.
Author Response
Comments 1: I appreciate your work, I believe the item treated is very relevant nowadays. I have some minor concerns: 1. paragraph 2.1, line 143 you wrote: PE and PET showed a frequency higher than 80%, 100%, and 82% respectively. You refer to 2 polymer but than report 3 values? Thank you for pointing this out. It's a typo. The misplaced comma has changed the meaning of the sentence. The sentence has been corrected. Line 170: “Two polymers showed a frequency higher than 80%, namely PE (100%) and PET (82%).”
2. paragraph 2.3, line 186 you stated that "it can be concluded the various particles in the samples analyzed were identified well". Was this conclusion derived from the spectra similarity index? Or do you use some other measuring method? This information is related to the values of spectral similarity. The sentence was rewritten for a better explanation (lines 217-219).
3. figure 4, the brown lines to which polymer refer? There isn't any indication in the label of the figure for this color.
Thank you for pointing this out. It was a mistake. The brown belongs to EPDM. The label colors were corrected in the revised manuscript (Figure 4).
4. In the conclusion, I would report at least a range (minimum and maximum values registered) of the number of particles detected with their errors. We agree with this comment. There is a new sentence in the conclusions (lines 400-402).
|

Round 2
Reviewer 2 Report
Comments and Suggestions for Authors
The major concerns raised in the initial review have been appropriately addressed, and the manuscript has shown clear improvements.
Author Response
Dear Reviewer
All questions raised in the manuscript have been answered and corrected where necessary.
In relation to the last revision, section 3.4 was missing and therefore reference 32 was missing. This section has been added to the manuscript.
Some corrections have been made to the references.
Thank you for your comments/corrections.
Best regards
Cristina
